# Retroviral Replicating Vector Toca 511 (*Vocimagene Amiretrorepvec*) for Prodrug Activator Gene Therapy of Lung Cancer

**DOI:** 10.3390/cancers14235820

**Published:** 2022-11-25

**Authors:** Hiroki Kushiya, Kei Hiraoka, Tomohiro Suzuki, Kazuho Inoko, Akihito Inagaki, Hiroki Niwa, Katsunori Sasaki, Toru Nakamura, Takahiro Tsuchikawa, Toshiaki Shichinohe, Douglas J. Jolly, Noriyuki Kasahara, Satoshi Hirano

**Affiliations:** 1Department of Gastroenterological Surgery II, Hokkaido University School of Medicine, Hokkaido 060-0808, Japan; 2Department of Clinical Research, National Hospital Organization (NHO) Hakodate National Hospital, Hokkaido 041-8512, Japan; 3Department of Neurological Surgery, University of California, San Francisco, CA 94158, USA; 4Tocagen Inc., San Diego, CA 92121, USA

**Keywords:** lung cancer, gene therapy, retrovirus vector

## Abstract

**Simple Summary:**

A unique gene therapy strategy based on tumor-selectively replicating retroviral vectors has shown promising results in clinical trials for glioma and gastrointestinal cancers. Here, we applied this strategy to the treatment of lung cancer, the leading cause of cancer deaths worldwide. Our results demonstrate that retroviral replicating vectors can achieve highly efficient delivery of reporter genes and prodrug activator genes to lung cancer cells in vitro and in vivo, with the latter enabling highly effective cancer cell killing upon exposure to prodrug, achieving significant therapeutic benefit in both subcutaneous and orthotopic pleural dissemination models of lung cancer. Retroviral replicating vector-based gene therapy thus represents a novel strategy with potential to address the unmet medical need for more effective ways to treat lung cancer.

**Abstract:**

Therapeutic efficacy of retroviral replicating vector (RRV)-mediated prodrug activator gene therapy has been demonstrated in a variety of tumor models, but clinical investigation of this approach has so far been restricted to glioma and gastrointestinal malignancies. In the present study, we evaluated replication kinetics, transduction efficiency, and therapeutic efficacy of RRV in experimental models of lung cancer. RRV delivering GFP as a reporter gene showed rapid viral replication in a panel of lung cancer cells in vitro, as well as robust intratumoral replication and high levels of tumor transduction in subcutaneous and orthotopic pleural dissemination models of lung cancer in vivo. Toca 511 (*vocimagene amiretrorepvec*), a clinical-stage RRV encoding optimized yeast cytosine deaminase (yCD) which converts the prodrug 5-fluorocytosine (5-FC) to the active drug 5-fluorouracil (5-FU), showed potent cytotoxicity in lung cancer cells upon exposure to 5-FC prodrug. In vivo, Toca 511 achieved significant tumor growth inhibition following 5-FC treatment in subcutaneous and orthotopic pleural dissemination models of lung cancer in both immunodeficient and immunocompetent hosts, resulting in significantly increased overall survival. This study demonstrates that RRV can serve as highly efficient vehicles for gene delivery to lung cancer, and indicates the translational potential of RRV-mediated prodrug activator gene therapy with Toca 511/5-FC as a novel therapeutic strategy for pulmonary malignancies.

## 1. Introduction

Lung cancer is the leading cause of cancer deaths worldwide, with over 2.2 million people diagnosed with lung cancer each year, and 1.8 million deaths [1]. Furthermore, around 40% of patients with non-small cell lung cancer (NSCLC) are diagnosed at stages III or IV [2,3]. In patients with pleural dissemination, the median survival time ranges from 6 to 9 months [2,4]. Treatment options for NSCLC with pleural dissemination are chemoradiotherapy, molecular targeted therapy, or immunotherapy [5]. However, the therapeutic goal for most patients with pleural dissemination is palliation [5,6,7], with patients experiencing dyspnea, malaise, and pain during cancer progression. Thus, novel therapeutic strategies are needed for patients with NSCLC, particularly those with advanced disease and pleural dissemination.

Gene therapy for lung cancer has not been widely used in clinical settings, although several clinical trials have been conducted [8]. In a Phase I trial evaluating replication-deficient adenovirus vector (Ad5CMV-p53) in lung cancer patients, treatment efficacy was limited. Despite the remarkably high titer of virus vector used, inadequate antitumor activity was a consequence of low transduction levels and limited distribution within the tumors after injection of conventional non-replicating vectors [9].

We have previously demonstrated that retroviral replicating vectors (RRVs) are capable of highly efficient replication in cancer cells in vitro, and highly selective and efficient tumor transduction in vivo. In addition to their intrinsic inability to infect post-mitotic cells, RRV have proven to be highly tumor-selective due to their intrinsic inability to infect post-mitotic cells, innate anti-retroviral immunity which is intact in normal cells but defective in cancer cells, and suppression of adaptive immunity in tumors. [10,11,12,13,14]. Prodrug activator gene therapy with RRV encoding yeast cytosine deaminase (yCD), which converts the prodrug 5-fluorocytosine (5-FC) to the chemotoxic drug 5-fluorouracil (5-FU), is highly effective in various tumor models [15,16,17,18,19,20]. Toca 511 (*vocimagene amiretrorepvec*), a clinical-stage RRV encoding an optimized yCD [21], has shown promising activity in multi-center Phase I dose escalation trials for glioma [22,23,24], including clinical improvement, radiographic complete responses, and prolonged survival compared to historical benchmarks. While an international Phase II/III trial in patients with recurrent high-grade glioma did not meet its endpoints overall [25], statistically significant differences in overall survival were observed with Toca 511/5-FC treatment in specific patient subgroups with IDH-mutant anaplastic astrocytoma, and further clinical investigation is on-going [25,26]. In addition, treatment responses and activation of anti-tumor immunity have recently been reported after intravenous delivery of Toca 511 followed by 5-FC treatment in patients with colorectal and pancreatic cancer [27]. To date, however, the therapeutic efficacy of RRV in vitro and in vivo has not been evaluated in lung cancer.

Prodrug conversion after Toca 511/5-FC treatment results in intratumoral generation of 5-FU, which exerts its antitumor effect by inhibiting DNA synthesis, and has been widely used for the treatment of a variety of cancers, including colorectal cancer, head and neck cancer, and breast cancer [28]. Precedent for use of 5-FU in patients with inoperable lung cancer dates back to the early 1960s [29]. Capecitabine (Xeloda), an oral prodrug that is converted to 5-FU by thymidine phosphorylase [30], has also been reported to show activity in lung cancer in case reports [31,32], and has been evaluated in combination with irinotecan for NSCLC [33], as well as in combination with temozolomide for advanced bronchopulmonary neuroendocrine tumors [34]. Tegafur is another prodrug that is converted to 5-FU by cytochrome 450 in the liver [35], and is used in combination with uracil, an inhibitor of dihydropyrimidine dehydrogenase, an enzyme that degrades 5-FU [36]. Several previous clinical studies of lung cancer have evaluated the effectiveness of uracil-tegafur (UFT), and adjuvant chemotherapy with UFT has been demonstrated to improve lung cancer patient survival [37]. In 2004, another oral fluoropyrimidine chemotherapeutic combination drug, S-1, which consists of tegafur, gimeracil (dihydropyrimidine dehydrogenase inhibitor), and oteracil (orotate phosphoribosyltransferase inhibitor), was approved in Japan as a monotherapy and under current guidelines is recommended for second-line treatment in patients with advanced NSCLC [38,39].

Hence, we hypothesized that Toca 511/5-FC treatment, which in many different cancer models has been shown to generate high intratumoral concentrations of 5-FU through local prodrug conversion directly within the tumor itself, might also elicit an enhanced therapeutic effect in lung cancer. In the present study, we first examined replication kinetics and transduction efficiency of RRV delivering the GFP reporter gene in preclinical models of lung cancer in vitro and in vivo, and subsequently evaluated the anti-cancer efficacy of Toca 511/5-FC treatment both in vitro and in vivo, using both subcutaneous tumor models as well as orthotopic models of pleurally disseminated lung cancer in both immunodeficient and immunocompetent hosts. Our findings indicate that RRV-mediated prodrug activator gene therapy shows significant potential for translational application to lung cancer.

## 2. Materials and Methods

### 2.1. Cell Lines and Culture 

Human lung adenocarcinoma cell line A549, human squamous cell carcinoma cell line H226, human small cell lung cancer cell line SBC-3, and mouse lung cancer cell line Ex-3LL were purchased from the Japanese Cancer Resources Bank (Ibaraki, Japan). Human embryonic kidney cells (293T) were purchased from the American Type Culture Collection (Manassas, VA, USA). Cell lines were cultured in either Dulbecco’s modified Eagle’s medium (DMEM) (A549, SBC-3, 293T) or Roswell Park Memorial Institute (RPMI) medium (H226, SBC-3, Ex-3LL), with 10% fetal bovine serum (Cell Culture Bioscience, Tokyo, Japan) and 1% penicillin–streptomycin (Life Technologies, Tokyo, Japan) at 37 °C in a 5% CO_2_ atmosphere. All media were purchased from WAKO (Tokyo, Japan). For in vivo imaging studies, A549 and Ex-3LL cells were transfected with pIRESpuro2-*luc*2 using ScreenFect™ A (Fujifilm WAKO Pure Chemical, Osaka, Japan) according to the manufacturer’s instructions. Subclones were then selected with 1 μg/mL puromycin (InvivoGen, San Diego, CA, USA). After four weeks of selection, stably transfected cells were harvested using a cloning ring (Takara Bio), subcultured at large scale, and then cryopreserved. The expression and activity of firefly luciferase in the cells of each clone (designated A549-luc2 and Ex-3LL-luc2, respectively) were tested using the IVIS imaging system (Xenogen/PerkinElmer, Waltham, MA, USA) after incubation with D-luciferin (Goryo Chemical, Sapporo, Japan).

### 2.2. Plasmid Constructs and RRV Production 

Plasmids pAC3-GFP (encoding RRV-GFP) and pAC3-yCD2 (encoding Toca 511) both contain an amphotropic murine retrovirus with an internal ribosome entry site (IRES)-transgene (GFP or optimized yCD, respectively) insertion, as described previously [21]. To produce RRV-GFP or Toca 511, 293T cells were transfected with pAC3-GFP or pAC3-yCD2 using Fugene HD Transfection Reagent (Promega, Madison, WI, USA). The cells were incubated in serum-free medium for 48 h before supernatant medium was collected and filtered through a 0.45 μM filter. Concentrated RRV was produced by Tocagen Inc. as previously described [17].

### 2.3. Replication Kinetics of RRV in Lung Cancer Cell Lines

Human and mouse lung cancer cell lines were infected with RRV-GFP at multiplicity of infection (MOI) 0.01 in the presence of 4 μg/mL polybrene (Sigma-Aldrich, St. Louis, MO, USA). Reverse transcriptase inhibitor 3′-azide-3′deoxythymidine (AZT, Sigma-Aldrich) was added on day 3 to block virus replication in negative control cells. GFP fluorescence of transduced cells was evaluated every 2–3 days by flow cytometry (FACSCanto II cytometer, BD Biosciences, San Jose, CA, USA) running FlowJo software v7.6.5 (TreeStar, Ashland, OR, USA). 

### 2.4. Cytotoxicity Assay In Vitro

Uninfected control cells, and cells infected with RRV-GFP control virus or Toca 511 prodrug activator virus, were cultured (1.0 × 10^3^ cells/well) in triplicate for 3–5 days with various concentrations of 5-FC. Cell viability was measured with a 3-(4-5-dimethylthiazol-2-yl)-5-(3-carboxymethoxyphenyl)-2-(4-sulfophenyl)-2H-tetrazolium salt (MTS) dye conversion assay (Promega). Optical absorbance of viable cells was determined relative to control wells without 5-FC. This experiment was performed three times.

### 2.5. Animal Studies 

All animal studies were approved by the Hokkaido University Animal Research Committee. Female 6- to 8-week-old BALB/c-nu/nu mice and C57BL/6 mice (CLEA, Tokyo, Japan) were housed under specific pathogen-free conditions.

### 2.6. Subcutaneous Tumor Models 

To evaluate RRV replication kinetics, subcutaneous tumor models were established first in immunodeficient mice, by injecting uninfected (99%) and RRV-GFP-infected (1%) human A549 cells (5 × 10^6^ cells/100 μL total volume) in Hanks’ Balanced Salt Solution (HBSS, Life Technologies, Carlsbad, CA, USA) into the right dorsal flanks of immunodeficient athymic nude (BALB/c-nu/nu) mice. Similarly, to establish immunocompetent subcutaneous tumor models, uninfected (99%) and RRV-GFP-infected (1%) murine Ex-3LL cells (5 × 10^6^ cells/100 μL total volume) in HBSS were injected into the right dorsal flanks of syngeneic C57BL/6 mice. On days 7, 14, and 21, tumor tissue was excised, and single-cell suspensions were prepared with the Tumor Dissociation Kit (human or mouse) and gentleMACS^TM^ Dissociator (Miltenyi Biotec, Bergisch Gladbach, Germany) according to the manufacturer’s protocol and evaluated for GFP expression by flow cytometry, as above.

For therapeutic efficacy studies in subcutaneous lung cancer models, A549 or Ex-3LL cell suspensions in HBSS were again injected into the right dorsal flanks of BALB/c-nu/nu mice or C57BL/6 mice, respectively, to establish subcutaneous tumors with RRV Toca 511 at initial infection levels of 1% as above, except for mice in control groups with uninfected tumors. All mice were randomized to 5-FC prodrug treatment or phosphate-buffered saline (PBS) vehicle control treatment, administered daily (500 mg/kg once per day) by intraperitoneal (i.p.) injection for 7 consecutive days at 7-day intervals, starting from day 14 or day 21 post-tumor establishment. Tumor volumes were calculated using the formula: volume = length × width^2^/2.

### 2.7. Pleural Dissemination Models 

Replicative kinetics and therapeutic efficacy were evaluated after intrathoracic RRV injection in a pre-established orthotopic pleural dissemination model of lung cancer. This model was established by injection of A549 cells (1.0 × 10^5^ cells/100 μL total volume) into the thoracic cavity of BALB/c-nu/nu mice. Two days after tumor establishment, RRV-GFP or RRV Toca 511 (4 × 10^6^ TU/mL concentration, 500 μL total volume) was administered via the same route. Disseminated pleural tumor masses were excised from mice injected with RRV-GFP on day 14 post-tumor establishment, and single-cell suspensions prepared using the human Tumor Dissociation Kit and the gentleMACS^TM^ Dissociator according to the manufacturer’s protocol. GFP expression levels were determined by flow cytometry using a FACSCanto II running FlowJo software V7.6.5. Mice injected with RRV Toca 511 were randomly assigned to groups receiving 5-FC prodrug or PBS vehicle control. As previously, 5-FC prodrug or PBS control treatments were administered daily (500 mg/kg once per day) by i.p. injection for 7 consecutive days at 7-day intervals starting from day 14 post-tumor establishment, and the percentage of surviving animals was examined in each group.

### 2.8. Bioluminescence Imaging 

Orthotopic pleural dissemination models of lung cancer were established as above by intrathoracic injection of A549-luc2 cells in BALB/c-nu/nu mice or Ex-3LL-luc2 cells in C57BL/6j mice, respectively, with Toca 511 infection started at 1% initial transduction (MOI equivalent 0.01) and 5-FC prodrug or PBS vehicle control treatments administered as above. Tumor burden was evaluated in real time at weekly or twice-weekly intervals by IVIS optical imaging (Xenogen/PerkinElmer) 10 min after injection of D-luciferin (150 mg/kg i.p.; GORYO Chemical, Sapporo, Japan), and imaging data analyzed with Living Image software v4.2 (Caliper, Hopkinton, MA, USA).

### 2.9. Analysis of RRV Systemic Biodistribution

Genomic DNA from multiple tissues (tumor, lung, heart, esophagus, liver, pancreas, rectum, kidney, ovary, spleen, bone marrow) after three daily 5-FC cycles following intrathoracic injection of RRV Toca 511 in the orthotopic pleural dissemination model, was purified using the DNeasy Blood & Tissue kit (Qiagen). qPCR experiments were performed in duplicate on a LightCycler^@^96 System (Roche Diagnostics) using TaqMan Universal PCR Master Mix II. Specific primers and probes were designed to target the 4070A amphotropic *env* gene (4070A-F, 5′-GCGGACCCGGACTTTTGA-3′; 4070A-R, 5′-ACCCCGACTTTACGGTATGC-3′; probe, FAM-CAGGGCACACGTAAAA-NFQ). Human *RNase P* (hRNase P, TaqMan Copy Number Reference Assay, Applied Biosystems) or mouse *Tfrc* (mouse, Tfrc, TaqMan Copy Number Reference Assay, Applied Biosystems) were used as internal control genes. A reference curve for RRV copy number was constructed using serial dilutions of pAC3-yCD2 in a background of gDNA from uninfected cells.

### 2.10. Statistical Analysis 

All analyses were performed using GraphPad Prism 8 software (GraphPad, La Jolla, CA, USA). Statistical comparisons were performed using Student’s *t*-test. Survival data were analyzed using the Kaplan–Meier method. In all analyses, *p*-values < 0.05 were considered statistically significant.

## 3. Results

### 3.1. Evaluation of RRV in Lung Cancer Cells In Vitro

#### 3.1.1. RRV Shows Rapid Viral Replication Resulting in High Levels of Cellular Transduction

To examine replication kinetics in vitro in human and murine lung cancer cells, we monitored the percentage of GFP-positive cells by flow cytometry every 2–3 days after inoculation with a retroviral replicating vector expressing GFP as a reporter gene (RRV-GFP) at a multiplicity of infection (MOI; i.e., virus: cell ratio) of 0.01, in order to initiate virus infection starting at an initial transduction level of 1%.

In all human lung cancer cell lines (A549, H226, SBC-3), the percentage of GFP-positive cells progressively increased over time, and reached 80% within 14 days after inoculation, and similarly, in murine lung cancer cell line Ex-3LL the percentage of GFP-positive cells reached 80% within 21 days after inoculation (Figure 1). In all cell lines, these increases in GFP-positive cells were inhibited (<5%) by adding the reverse transcriptase inhibitor drug azidothymidine (AZT), indicating that active virus replication was required for efficient gene transfer. The cytofluorimetric data in A549 were shown in Appendix A.

#### 3.1.2. Toca 511/5-FC Treatment Shows Remarkable Toxicity to Lung Cancer Cells In Vitro

Cytotoxicity induced in Toca 511 (RRV-yCD)-transduced lung cancer cells after exposure to several concentrations of 5-FC prodrug was examined by MTS assay. Uninfected lung cancer cells (naive) and RRV-GFP-transduced cells were used as controls (Figure 2). In both uninfected and RRV-GFP infected control A549 human lung cancer cells, no significant growth inhibition was observed after incubation with 5-FC prodrug at concentrations up to 1 mM (Figure 2A). In contrast, Toca 511-transduced A549 cells showed over 60% growth inhibition after exposure to 0.1 mM 5-FC, and over 90% growth inhibition after exposure to 1.0 mM 5-FC (Figure 2A). While uninfected and RRV-GFP infected control Ex-3LL murine lung cancer cells showed non-specific growth inhibition after exposure to 0.1 mM 5-FC, this was not observed after exposure to 0.01 mM 5-FC (Figure 2B). In contrast, Toca 511-transduced Ex-3LL cells showed over 80% growth inhibition after exposure to 0.01 mM 5-FC (Figure 2B). These results indicate that expression of the human codon-optimized yeast cytosine deaminase transgene in Toca 511-infected lung cancer cells enabled intracellular prodrug conversion resulting in cytotoxicity.

### 3.2. Evaluation of RRV-GFP Tumor Transduction and Toca 511/5-FC Therapeutic Efficacy in Subcutaneous Lung Cancer Models In Vivo

#### 3.2.1. RRV Shows Significant Viral Replication and Tumor Transduction in Both Immunodeficient and Immunocompetent Subcutaneous Tumor Models of Lung Cancer In Vivo

Intratumoral RRV replication kinetics and resultant tumor transduction levels over time were evaluated, first using subcutaneous tumor models of A549 human lung cancer xenografts established in immunodeficient athymic nude mice, by flow cytometry of disggregated tumor cells at serial time points after tumor establishment. Starting from an initial RRV-GFP transduction level of 1%, the percentage of GFP-positive cells on days 7, 14, and 21 post-tumor establishment was 39.5 ± 7.9%, 78.1 ± 2.1%, and 82.7 ± 4.4% (Figure 3A). 

Next, subcutaneous tumor models of murine Ex-3LL lung cancer were established in immunocompetent syngeneic mice. Again starting from an initial RRV-GFP transduction level of 1%, viral replication was observed to be slower than in the A549 model, but significant tumor transduction levels were achieved, with excised tumor cells positive for GFP expression at progressively increasing levels of 16.3 ± 5.9%, 21.5 ± 3.5%, and 51.1 ± 7.1% on days 7, 14, and 21 post-tumor establishment, respectively (Figure 3B).

#### 3.2.2. Toca 511/5-FC Treatment Achieves Significant Anti-Tumor Efficacy in Both Immunodeficient and Immunocompetent Subcutaneous Tumor Models of Lung Cancer

For initial studies to evaluate the in vivo efficacy of Toca 511/5-FC treatment for lung cancer, we again established subcutaneous tumor models of A549 human lung cancer in immunodeficient athymic BALB/c nude mice, and Ex-3LL murine lung cancer in immunocompetent syngeneic C57BL/6 mice. Both models were established as untransduced control tumors or infected with Toca 511 at an initial transduction level of 1%. Both groups in each model were then randomized to receiving either 2–3 cycles of 5-FC prodrug treatment, or phosphate-buffered saline (PBS) vehicle as a control treatment. In both models (A549 and Ex-3LL), untransduced tumors showed no significant growth inhibition whether saline or 5-FC was administered, and Toca 511-transduced tumors showed no significant growth inhibition after PBS vehicle treatment (Figure 4). In contrast, Toca 511-transduced subcutaneous tumors treated with 5-FC showed highly significant growth inhibition compared to PBS-treated controls (*p* = 0.0051 in A549 model (Figure 4A), *p* = 0.0018 in Ex-3LL model (Figure 4B). Tumor growth inhibition was particularly remarkable in the Ex-3LL tumor model, in which 4 out of 5 mice showed complete tumor regression.

#### 3.2.3. Minimal Systemic Biodistribution of RRV in Normal Tissues after Toca 511/5-FC Treatment in Subcutaneous Tumor Models of Lung Cancer

Genomic DNA was extracted from various tissues after Toca 511/5-FC treatment in the above A549 and Ex-3LL subcutaneous tumor models, and systemic biodistribution of RRV was evaluated by quantitative PCR to detect integrated vector sequences. As expected, in subcutaneous A549 human lung cancer xenograft models established in immunodeficient athymic nude mice, RRV signals were evident primarily in spleen and bone marrow (4.32 ± 3.15 and 12.64 ± 0.42 vector copies/100 cell genomes) (Figure 5A). Other tissues showed minimal RRV signals, which could be attributable to residual peripheral blood in the harvested tissue specimens. Notably, however, RRV signals were not observed in spleen or bone marrow from subcutaneous Ex-3LL murine lung cancer models established in immunocompetent syngeneic mice, which overall showed minimal systemic RRV biodistribution across all tissues examined (Figure 5B). This indicates that an intact immune system is capable of controlling virus replication in systemic normal tissues.

### 3.3. Evaluation of RRV in Orthotopic Lung Cancer Pleural Dissemination Models In Vivo

#### 3.3.1. RRV Shows Significant Tumor Transduction after Intrathoracic Injection in an Orthotopic Pleural Dissemination Model of Human Lung Cancer 

The efficiency of RRV-mediated tumor transduction was also evaluated in an orthotopic model of non-resectable human lung cancer with pleural dissemination. Pleurally disseminated tumors were generated by intrathoracic injection of A549 human lung cancer cells in athymic nude mice on day0, and RRV-GFP was then injected via the same route into the thoracic cavity on day2. Subsequently, the percentage of GFP-positive cells was evaluated by flow cytometric analysis of disaggregated tumor cells. By day 14 post-tumor establishment, tumor cells were 65.8 ± 4.2% positive for GFP expression by flow cytometry (Figure 6A). Thus, significant levels of tumor transduction could be achieved after intrathoracic injection of RRV in this orthotopic pleural dissemination model.

#### 3.3.2. Toca 511/5-FC Treatment Improved Overall Survival in an Orthotopic Pleural Dissemination Model of Human Lung Cancer

Therapeutic efficacy of Toca 511/5-FC gene therapy was then evaluated in this orthotopic pleural dissemination model of human lung cancer. After establishment of disseminated pleural tumors as above, all animals received intrathoracic injection of Toca 511 via the same route, then were randomized to groups treated with either 5-FC prodrug or PBS vehicle control daily days for 7-day intervals every other week, starting from day 14 post-tumor establishment. In the PBS control group, median overall survival time was 64 days, with none surviving longer than 77 days. In contrast, the 5-FC prodrug treatment group showed significantly prolonged survival, with a median overall survival of 99 days (*p* = 0.0138; hazard ratio 0.30, 95% CI: 0.078–1.1) (Figure 6B).

#### 3.3.3. Minimal Systemic Biodistribution of RRV after Toca 511/5-FC Treatment in Orthotopic Pleural Dissemination Model of Human Lung Cancer

Systemic biodistribution of RRV after intrathoracic injection in the above human lung cancer pleural dissemination model was examined by qPCR of genomic DNA. As expected, high levels of RRV signal were observed in tumor tissues, while only low levels of RRV were detected in normal tissues (Figure 7). As previously, in this immunodeficient lung cancer model, slightly elevated signals were seen in bone marrow and spleen (2.81 ± 2.75 and 1.29 ± 0.21 vector copies/100 cell genome equivalents of DNA, respectively) and this time, also in lung (3.91 ± 2.33 vector copies/100 genome equivalents) after intrathoracic injection, but values were <1 in all other tissues examined.

#### 3.3.4. Tumor Growth Inhibition over Multiple Prodrug Treatment Cycles Visualized in Real Time by Bioluminescence Imaging after Toca 511/5-FC Treatment in Orthotopic Lung Cancer Pleural Dissemination Models

Bioluminescence emitted by human A549 and murine Ex-3LL lung cancer cells stably transfected with the firefly luciferase reporter gene, designated as A549-luc2 and Ex-3LL-luc2, respectively, was evaluated by optical imaging in vitro after incubation with luciferin. A proportional relationship between cell numbers and luminescent signal intensity was established (Appendix A).

Orthotopic pleural dissemination models of A549-luc2 human lung cancer in immunodeficient athymic nude mice as well as Ex-3LL-luc2 murine lung cancer in immunocompetent syngeneic mice were established as above. In this study, the initial Toca 511 transduction level was adjusted to start infection at 1% (i.e., MOI equivalent 0.01) in both models. All animals were then randomized to 7-day cycles of either prodrug (5-FC) or vehicle control (PBS) treatments every other week, starting from day 14 in the A549-luc2 model or day 7 in the Ex-3LL-luc2 model, respectively. Tumor burden was monitored in real time by optical imaging of tumor bioluminescence. In both models, the PBS vehicle control groups showed tumor progression, as indicated by increasing luminescent signal intensity over time (Figure 8). In contrast, 5-FC treatment groups in both models showed tumor growth inhibition or stable disease, as indicated by decreasing luminescent signal intensity coinciding with the first 5-FC prodrug cycle in the A549-luc2 model (Figure 8A) and during or immediately following each cycle in the Ex-3LL-luc2 model (Figure 8B). Overall, Toca 511/5-FC treatment was associated with significant reductions in mean signal intensity in both the A549-luc2 model (*p* = 0.0147) and Ex-3LL-luc2 model (*p* = 0.0326) when compared with their respective control groups at the last timepoint when control animals were all still alive (Figure 8).

As expected based on these optical imaging results (and consistent with previous survival data from the orthotopic pleural dissemination model generated with parental A549 cells as shown above), in this study, overall survival was significantly prolonged in the A549-luc2 orthotopic pleural dissemination model after Toca 511/5-FC treatment, with a median overall survival time of 94 days, as compared to controls also with Toca 511-infected tumors but treated with PBS vehicle, which resulted in a median overall survival time of 64 days (*p* = 0.0027; hazard ratio 0.19, 95% CI: 0.036–0.95) (Figure 9A).

Tumor growth in the orthotopic pleural dissemination model of murine Ex-3LL-luc2 lung cancer in immunocompetent syngeneic mice appeared to be more aggressive compared to the A549 and A549-luc2 models, as optical imaging of tumor bioluminescence indicated rapid tumor growth in the control group with Toca 511-infected tumors but treated with PBS vehicle, and this was associated with a shorter median overall survival time of 23 days. Rapid recurrence of tumor bioluminescence signals was also observed in between prodrug cycles during Toca 511/5-FC treatment, although as noted, reductions in tumor signals were observed coinciding with prodrug treatment, and mean signal intensity was significantly reduced overall as compared to PBS controls (Figure 8B). Consistent with these observations, there was statistically significant prolongation of median overall survival to 30 days after Toca 511/5-FC treatment in the orthotopic Ex-3LL-luc2 pleural dissemination model as compared to PBS controls, and 20% of treated mice showed longer term survival (*p* = 0.0049; hazard ratio 0.31, 95% CI: 0.098–0.98) (Figure 9B).

## 4. Discussion

RRV can efficiently replicate in cancer cells and integrate into the cancer cell genome, so that each infected cancer cell is, in turn, stably converted into a virus producer cell capable of producing further progeny virus. Thus, even if only a small number of tumor cells are infected initially, RRV could then spread to surrounding noninfected tumor cells, achieving in situ amplification of the input dose. In previous studies, RRV prodrug activator gene therapy has shown remarkable antitumor efficacy in a wide variety of cancer models [15,16,17,18,19,20], but this approach had not previously been evaluated for application to lung cancer. This represents the first preclinical study to demonstrate that RRV-mediated prodrug activator gene therapy could be efficacious in lung cancer.

In the present study, we found that RRV rapidly replicates in lung cancer cells in vivo and in vitro, even after initial virus infection at an MOI of 0.01, i.e., starting from a transduction level of 1%, and achieved significant growth inhibition in subcutaneous tumor models of lung cancer after Toca 511/5-FC treatment. We further demonstrated efficient transduction by RRV in murine orthotopic models of experimental lung cancer, which mimics pleural dissemination observed in patients with advanced disease. In this model, we found that RRV was capable of selective, highly efficient replicative spread following intrathoracic injection, demonstrating the possibility of intrathoracic RRV administration in a clinical setting using a thoracic catheter. As video-assisted thoracoscopic surgery is widely performed [40,41], direct injection of Toca 511 into pleural nodules could be easily performed during thoracoscopic surgery.

Although efficient RRV transduction was observed in the lung cancer models tested (Figure 4 and Figure 7), it is possible that some of the viral genomes were mutated by cellular restriction factors, such as APOBEC cytidine deaminases, in the tumor samples. A recent report identified numerous G to A mutations that may convert a tryptophan codon to a stop codon in integrated Toca 511 genomes when evaluating patient tumor samples [24]. Of these, the most common was identified in the yCD2 region, which presumably eliminated functional yCD2 proteins. However, this same paper also reported that yCD2 protein expression was detected even where deleterious mutations were observed at high frequency in resected tumor tissues from patients administered Toca 511 and multiple rounds of 5-FC. This suggests that a reservoir of functional viral particles remains regardless of the number of mutations induced by the APOBECs. Although this result cannot be applied directly to our study given the species difference, we assume that functional virus reservoirs persist even after the endpoint of the in vivo study described. In addition, the majority of tumor samples analyzed for mutations showed at least 10% functional yCD, which represents the threshold for achieving an effective bystander effect that can eliminate an entire population of cancer cells [42]. Furthermore, in immunocompetent hosts, a “distant bystander effect” of immunological activation resulting in anti-tumor immune responses also contribute to therapeutic benefit [43]. In fact, the immunocompetent Ex-3LL model (Figure 8B) and our previous study, which used a different cancer model [18], demonstrated a clear therapeutic effect in response to 5-FC cycling, suggesting that functional Toca 511 reservoirs continue to infect tumor cells as they grow, and bystander effect induce antitumor effects.

In this study, the safety of Toca 511/5-FC treatment was demonstrated by the limited systemic biodistribution of RRV. In our preclinical models, RRV copy numbers were minimal in most tissues, except for the tumor, indicating that RRV can selectively transduce lung cancer, consistent with previous studies in other cancer models in which RRV copy numbers were at almost undetectable levels in all tissues [20]. Furthermore, systemic biodistribution of RRV may be further restricted in immunocompetent hosts as compared to immunodeficient hosts. Both innate anti-retroviral host defenses such as APOBEC3, DDX41 [44,45], as well as adaptive immunity [46], restrict retrovirus infection. Indeed, preclinical safety studies of amphotropic murine retrovirus in non-human primates supported initiation of the very first clinical trials of human gene therapy in 1990 [47]. In addition, in first-in-human clinical trials for recurrent high-grade glioma, Toca 511/5-FC treatment demonstrated a better safety profile with fewer severe toxicities than lomustine [22]. Together, these results demonstrate the excellent safety profile of RRV. As a further safety mechanism, even if normal cells were to be infected with RRV, administration of a reverse transcriptase inhibitor such as azidothymidine would restrict further RRV infection.

The time course of RRV replication in the A549 human lung cancer model xenografted in immunodeficient mice was similar to that of RRV replication observed in this cell line in vitro. In contrast, in the Ex-3LL murine lung cancer model established in syngeneic immunocompetent mice, RRV replication was slower than observed in vitro. As noted, one possible explanation is that virus replication was hindered by innate and adaptive anti-viral immune responses, which are active in immunocompetent C57BL/6 mice, while athymic BALB/c nude mice have defects in both innate anti-retroviral defenses and lack T-cell mediated adaptive immunity [48,49,50,51]. A second possible explanation is the highly aggressive nature of Ex-3LL tumor growth, which was faster than that in the A549 model. In the Ex-3LL model, subcutaneous tumor sizes reached around 1000 mm^3^ by day 42. It is possible that Ex-3LL cell growth outpaced the production of RRV progeny virus, resulting in a slower increase in the percentage of RRV-infected cells.

Despite differences in rates of virus replication and tumor transduction, significant anti-tumor effects of Toca 511/5-FC treatment were observed in both models. In the immunocompetent Ex-3LL syngeneic model, despite the lower rate of infection in vivo, complete responses were observed in several mice after administration of 5-FC. This suggests that, separate from the direct cytotoxic effect of intratumoral 5-FU and its “bystander effect”, anti-tumor immune responses may also be activated after Toca 511/5-FC treatment. Hiraoka et al. [18] previously demonstrated that mice with intracranial gliomas previously treated with Toca 511/5-FC rejected subcutaneous re-challenge with uninfected glioma cells, indicating that a systemic immunological response against tumor antigens had been induced. Mitchell et al. [52] demonstrated that after Toca 511/5-FC treatment, myeloid-derived suppressor cells (MDSC) were significantly decreased in subcutaneous tumors, while CD4+ and CD8+ T cells were both significantly increased, suggesting that “bystander effects” due to intratumoral production of 5-FU resulted in local myelotoxicity within the tumor microenvironment, while maintaining systemic immune function. In recent years, cancer immunotherapy, including immune checkpoint inhibitors, has become established for treatment of NSCLC [53,54,55]. However, PD-1 pathway blocking drugs have an average response rate of only 20–30% across different tumor types, and immune-related adverse events contribute to treatment discontinuation [56,57]. We propose that tumor mass reduction and tumor antigen release by Toca 511/5-FC treatment may enhance the efficacy of immune checkpoint blockade without increasing systemic toxicity, although further investigations involving such combination regimens are needed.

Many patients with advanced lung cancer have symptoms such as pain, dyspnea, cough, decreased appetite, and depression related to cancer progression [58]. Because of the possibility of adverse events and the need to maintain quality of life, there is no indication for chemotherapy in majority of these patients. However, considering the safety profile described above, Toca 511/5-FC treatment could be useful in these patients. In a clinical setting, even if complete responses are not achieved, Toca 511/5-FC treatment may improve quality of life. Currently, multidisciplinary treatment combining surgery, chemotherapy, radiotherapy, and immune checkpoint inhibitors are widely used for advanced lung cancer [59,60,61]. Thus, further investigation of Toca 511/5-FC as a new treatment modality to improve survival in patients with advanced lung cancer is warranted.

## 5. Conclusions

Our results presented here demonstrate the efficient replication kinetics and transduction of RRV in several models of lung cancer. Moreover, in a preclinical orthotopic pleural dissemination model of lung cancer, survival was significantly improved following intrathoracic RRV injection and systemic administration of 5-FU. In conclusion, Toca 511/5-FC treatment may be a promising therapeutic strategy in lung cancer.

## Figures and Tables

**Figure 1 cancers-14-05820-f001:**
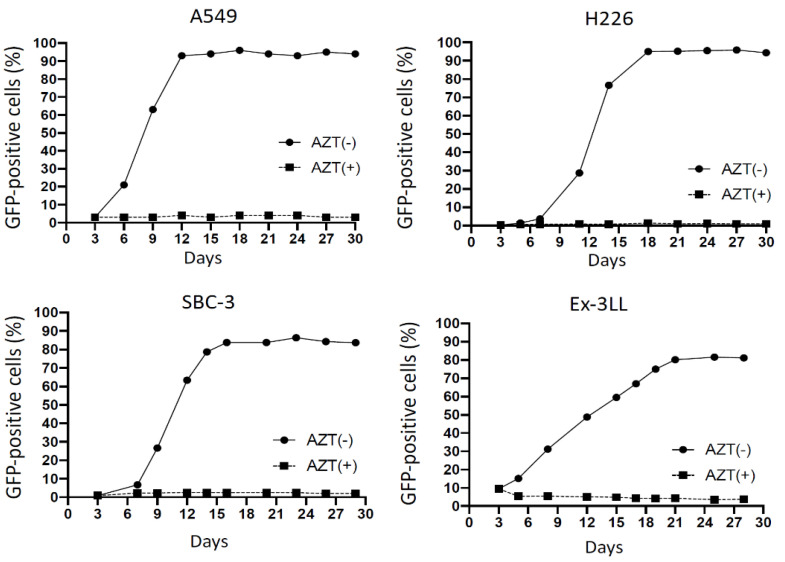
Replication kinetics of RRV-GFP in lung cancer cell lines in vitro. On day 0, all cell lines were transduced with RRV-GFP at MOI 0.01. In negative control cells, reverse transcriptase inhibitor AZT was added on day 3. Cells were evaluated for GFP by flow cytometry every 2–3 days.

**Figure 2 cancers-14-05820-f002:**
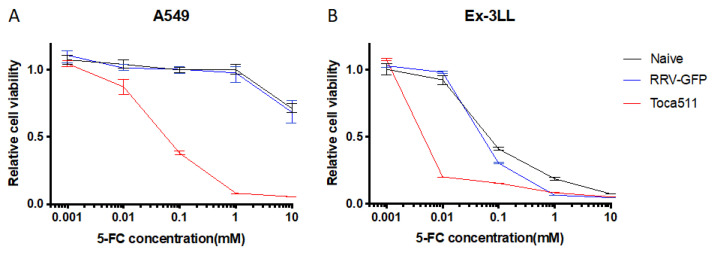
Cytotoxicity after 5-FC prodrug treatment in uninfected (naïve), RRV-GFP-infected, and Toca 511-infected A549 (**A**) and Ex-3LL cells (**B**) in vitro. Cell viability was measured by MTS assay after incubation with different concentrations of 5-FC, as shown, for 4 days. Data are reported as the percentage (mean ± SD) of viable cells, normalized to uninfected control cells without prodrug.

**Figure 3 cancers-14-05820-f003:**
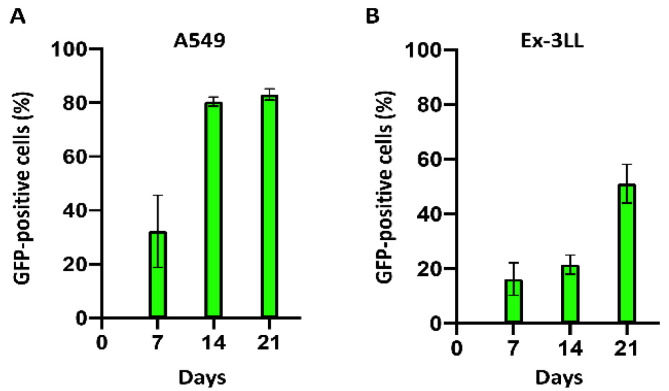
Spread of RRV-GFP in subcutaneous tumor models of lung cancer. Subcutaneous A549 human lung cancer models in immunodeficient nude mice (**A**) and Ex-3LL murine lung cancer models in syngeneic immunocompetent mice (**B**) were established on day 0 and RRV-GFP infection initiated at a starting level of 1% (*n* = 3 per group). Tumors from replicate cohorts were excised on the indicated days, and GFP fluorescence of disaggregated tumor cells was analyzed by flow cytometry.

**Figure 4 cancers-14-05820-f004:**
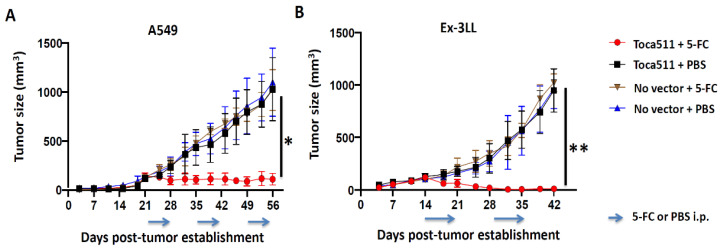
Therapeutic efficacy of Toca 511 and 5-FC therapy in subcutaneous tumor models of A549 human lung cancer and Ex-3LL murine lung cancer. Dorsal flank tumors were established and Toca 511 infection was initiated starting from an initial transduction level of 1% (*n* = 5/group). Uninfected tumors served as negative controls (‘No vector’). Data are shown as the mean ± SD. (**A**) Tumor growth curves for subcutaneous A549 human lung cancer xenograft models. Prodrug (5-FC) or saline vehicle (PBS) treatments were administered daily (500 mg/kg i.p.) for 7 consecutive days every other week, starting from day 21 post-tumor establishment (indicated by arrows); * *p* = 0.0051. (**B**) Tumor growth curves for subcutaneous Ex-3LL murine lung cancer tumor models in immunocompetent syngeneic mice. As above, prodrug (5-FC) or saline vehicle (PBS) treatments were administered daily (500 mg/kg i.p.) for 7 consecutive days every other week, starting from day 14 post-tumor establishment (arrows); ** *p* = 0.0018.

**Figure 5 cancers-14-05820-f005:**
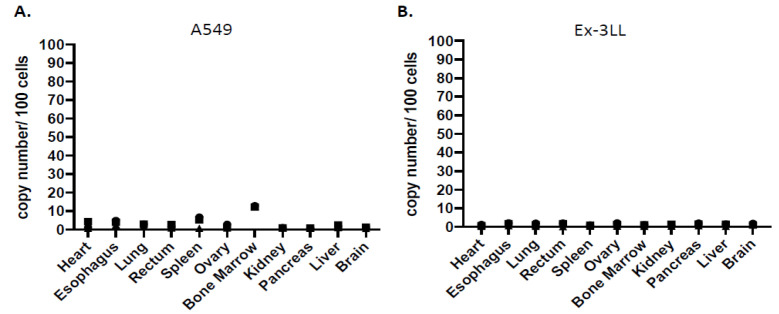
Systemic biodistribution of Toca 511 in subcutaneous lung cancer models. Genomic DNA was extracted from various tissues (*n* = 3 per tissue) after Toca 511/5-FC-treatment in (**A**) immunodeficient A549 human lung cancer xenograft models and (**B**) immunocompetent Ex-3LL murine lung cancer models. Integrated RRV sequences were quantitated by qPCR based on reference curve values and expressed as vector copy numbers per 100 cell genome equivalents of DNA. The logarithmic graph was shown in Appendix A.

**Figure 6 cancers-14-05820-f006:**
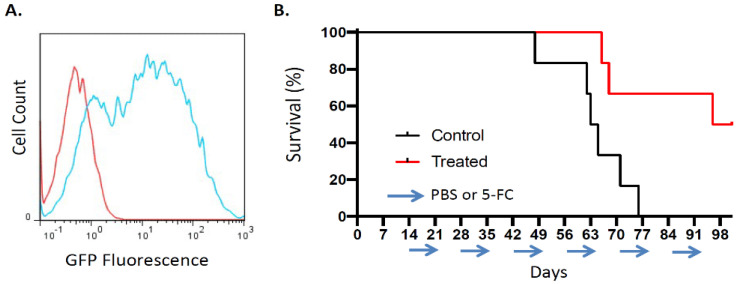
Tumor transduction and therapeutic efficacy of RRV in orthotopic pleural dissemination model of human lung cancer after vector injection directly into the thoracic cavity. Disseminated pleural tumors were established by intrathoracic injection of A549 cells into athymic nude mice on day 0 (*n* = 6). (**A**) RRV-GFP vector (2 × 10^6^ TU/500 μL) was injected into the thoracic cavity on day 2, and tumors were excised on day 14 and disaggregated for flow cytometry. Histograms show representative flow cytometric results quantitating GFP expression levels in uninfected negative control tumor (red curve) and in RRV-GFP-infected tumor (blue curve, 71.1% GFP-positive). (**B**) Kaplan–Meier survival analysis in lung cancer pleural dissemination model established as above, after intrathoracic injection of RRV Toca 511 via the same route, followed by 5-FC prodrug (‘Treated’) or PBS vehicle (‘Control’) administered in 7-day cycles every other week (arrows).

**Figure 7 cancers-14-05820-f007:**
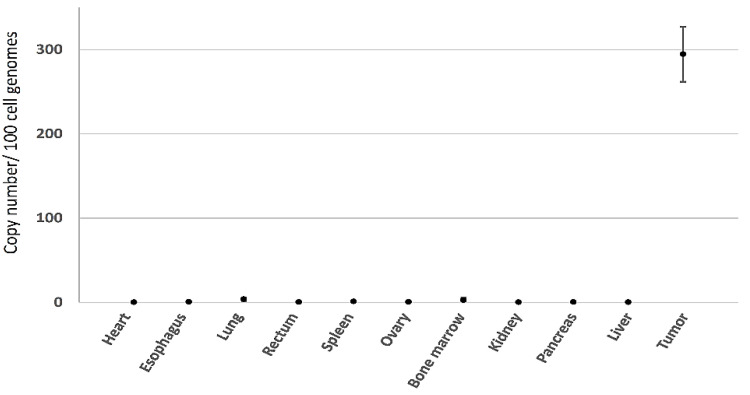
Systemic biodistribution of Toca 511 in orthotopic human lung cancer model. Genomic DNA was extracted from various tissues after Toca 511/5-FC-treatment in pleural dissemination model of A549 human lung cancer in athymic nude mice, as described in Figure 6B (*n* = 3 per tissue). Integrated RRV sequences were quantitated by qPCR based on reference curve values and expressed as vector copy number per 100 cell genome equivalents of DNA. The logarithmic graph was shown in Appendix A.

**Figure 8 cancers-14-05820-f008:**
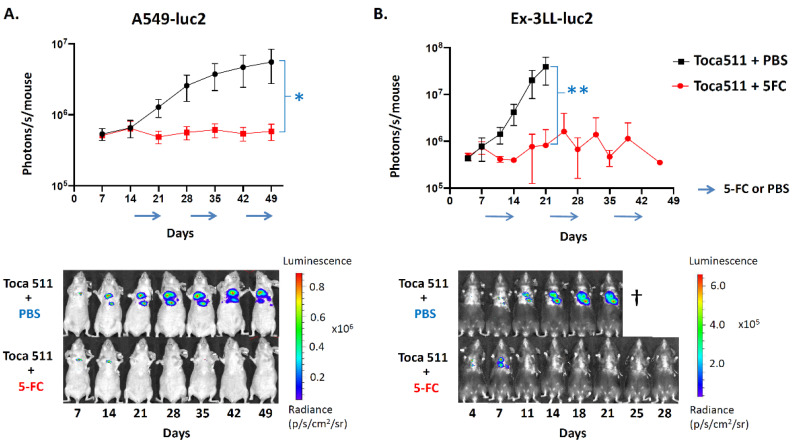
Bioluminescence imaging of Toca 511/5-FC prodrug activator gene therapy in orthotopic pleural dissemination models of lung cancer. Luciferase-marked human A549-luc2 (**A**) or murine Ex-3LL-luc2 (**B**) tumors were established by intrathoracic injection and Toca 511 infection as described in Methods, and IVIS optical imaging was performed at serial time points (*n* = 5 per group). (**A**) In the A549-luc2 model, 7-day cycles of 5-FC (500 mg/kg, red line) or PBS (black line) were administered every other week (arrows) from day 14 post-tumor establishment; * *p* = 0.0147. (**B**) In the Ex-3LL-luc2 model, 7-day cycles of 5-FC (500 mg/kg, red line) or PBS (black line) were administered every other week starting from day 7; ** *p* = 0.0326. Lower panels show representative serial imaging results from an individual animal in each treatment group for each model.

**Figure 9 cancers-14-05820-f009:**
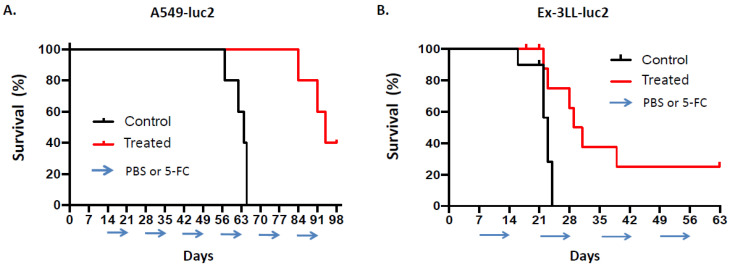
Overall survival after Toca 511/5-FC treatment in orthotopic models of human xenograft and murine syngeneic lung cancer. Pleural dissemination models of human A549-luc2 (**A**) or murine Ex-3LL-luc2 (**B**) lung cancer were established and Toca 511/5-FC treatment was performed as described in Figure 8 and in Methods. (**A**) Kaplan–Meier survival analysis in the A549-luc2 model (*n* = 5 per group): PBS vehicle (‘Control’) or 5-FC prodrug (‘Treated’) were administered daily in 7-day cycles every other week, starting from day 14 post-tumor establishment (cycles indicated by arrows). (**B**) Kaplan–Meier survival analysis in the Ex-3LL-luc2 model (*n* = 10 per group): PBS vehicle (‘Control’) or 5-FC prodrug (‘Treated’) were administered daily in 7-day cycles every other week, starting from day 7 post-tumor establishment (arrows).

## Data Availability

The data presented in this study are available on request from the corresponding author.

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
