# Peer review of "Retroviral Replicating Vector Toca 511 (Vocimagene Amiretrorepvec) for Prodrug Activator Gene Therapy of Lung Cancer"

_cancers, 2022, doi:10.3390/cancers14235820_

Round 1

Reviewer 1 Report (Previous Reviewer 3)

regarding my first comment in the previous review (Using qPCR analysis, the authors demonstrated that tumor masses of treated mice maintained a high level of viral copy number in the genome. These data do not demonstrate whether viral particles are still infectious and sensitive to 5-FC. It is known from the literature that Toca511 can undergo mutations during in vivo spreading, which can generate nonfunctional viral genomes or inactivate the cytosine deaminase gene (PMID:29945998). Therefore, even considering that no mice show complete regression in orthotopic models, the authors should analyze whether, at the experimental endpoints, tumor masses still contain replicating and infectious viral particles, and whether infected cells remain sensitive to 5-FC treatment. Moreover, they could analyze whether viral particles spread evenly throughout the entire tumor mass, reaching the migrating tumor cells.) I would have expected an experiment and not just only example of experiments performed in different context. However, I ask to the authors to at least discuss these points in the discussion section.

Author Response

Thank you for your kind comment.

We have added sentence to the text as suggested.(P12 L489- L510)

Reviewer 2 Report (Previous Reviewer 2)

The authors addressed my concerns.

Author Response

Thank you for your kind comment.

Reviewer 3 Report (Previous Reviewer 1)

This manuscript is good, and I only have one concern:

Could the RRV infect the lung cells specifically? If not, how to ensure the targeting effeciency?

Author Response

We have added sentence to the text as suggested.(P2 L71- L74) RRV infects a wide variety of cancer cells as shown in previous studies, meaning that the infectivity is not “specific” to lung cancer cells. On the other hand, the tumor “selectivity” of RRV infection is remarkable as seen in Fig. 7 where there was a stark difference in the virus copy number detected between the tumor samples and the normal lung tissue following intrathoracic injection in the immunodeficient animal model. Furthermore, in an immunocompetent animal, RRV replication is highly restricted by various innate anti-retroviral immunity systems, such as APOBEC3 and DDX41, as well as adaptive immunity in normal tissues (Stavrou S, et al., mBio, 2018, Takeuchi H, et al., Microbiol Immunol, 2008). In a clinical setting, as described in the study  carried out by Hogan, D. J. et al., Clin Can Res, 2018, it would be useful to analyze RRV copy number by PCR and its functional yCD2 protein by immunohistochemical staining both in tumor samples and adjacent and distant normal lung tissues to ensure the targeting efficiency within a thoracic cavity.

This manuscript is a resubmission of an earlier submission. The following is a list of the peer review reports and author responses from that submission.

Round 1

Reviewer 1 Report

In this work, Hiroki et al demonstrated the efficient replication kinetics and transduction of RRV in several models of lung cancer. Moreover, in a preclinical orthotopic pleural dissemination model of lung cancer, survival was significantly improved following intrathoracic RsRV injection and systemic administration of 5-FU. The data presented are generally strong, and appear convincing. The work is based on a very strong foundation with clear and robust data.

Just curious, any possibility of adverse events of using RsRV? Does this method affect the normal tissue cells, such cell viability or cell cycle? What's the advantage of this method, because in the mouse model, so many approaches are effective. The author should discuss or confirm more to convince the readers.

Minor Comment:

Some language errors need to be correct in this manuscript. Such as “in vivo” and “in vitro” should be italicized.

Reviewer 2 Report

The authors examined replication kinetics, transduction efficiency, and therapeutic efficacy of RRV-mediated prodrug activator gene therapy in both cell culture and animal models. Findings from the present study support that RRV vectors can efficiently replicate and achieve significant levels in tumors. The authors concluded that RRV-mediated prodrug activator gene therapy with Toca 511/5-FC could be a therapeutic approach to improve chemotherapy (5-FU) in lung cancer.  This manuscript provided interesting data and new findings on the role of RRV in cancer therapy. However, there are some weaknesses and concerns that need to be carefully addressed, especially regarding immune response, and off-target effect:

1.   The authors examined the impact of treatment in immunodeficient athymic BALB/c nude mice, and immunocompetent syngeneic C57BL/6 mice (Figure 4). In immunodeficient athymic BALB/c nude mice, treatment started from day 21 post-tumor establishment. In contrast, immunocompetent syngeneic C57BL/6 mice received treatment after day 14 post-tumor establishment. Clearly, the presence of immune cells could impact tumor establishment in vivo. Additionally, RRV vectors are foreign antigens in immunocompetent mice, potentially hindering tumor growth and confounding therapeutic responses. In Figure 5, the authors concluded that “an intact immune system is capable of controlling virus replication in systemic normal tissues”. For these reasons, this reviewer suggests IHC characterizing tumor infiltering immune cells (e.g., macrophages and NK cells), which would strengthen the current version of the manuscript.

2.     Authors concluded that RRV-mediated prodrug activator gene therapy can efficiently improve the antitumor activity of 5-FU in mouse models. Figure 7 (systemic biodistribution of Toca 511 in orthotopic human lung cancer model) shows that “high levels of RRV signal were observed in tumor tissues, while only low levels of RRV were detected in normal tissues”.  Authors need to explain why RRV selectively spread to tumor cells in vivo.

Reviewer 3 Report

In the manuscript by Kushiya et al. entitled "Retroviral Replicating Vector Toca 511 (vocimagene amiretrorepvec) for Prodrug Activator Gene Therapy of Lung Cancer" authors described the therapeutic efficacy of the retrovirus Toca511 expressing yeast cytosine deaminase, that can convert the prodrug 5-FC in the cytotoxic 5-FU molecule, in lung cancer cell lines both in vitro and in vivo. As also stated by the authors, this treatment has already been proven safe and potentially useful in preclinical studies for the treatment of different solid tumors and went through a Phase II/III clinical trial for glioblastomas.

The main general limitation of the study is the use of cancer cell lines instead of patient derived cells that could have a different response to treatment, but I think the former may be useful for an initial analysis of therapeutic efficacy as done by the authors in the manuscript.

As a whole, the manuscript is well written, the research design is appropriate, the conclusions are supported by the results and it could be a contribution to the journal. However, authors should address the following points:

-          By qPCR analys, authors demonstrated that tumor masses of treated mice maintain a high level of viral copy number in the genome. These data do not demonstrate whether viral particles are still infectious and sensible to 5-FC. It is known from literature, that Toca511 can undergo mutations during in vivo spreading that can generate nonfunctional viral genomes or inactivate the cytosine deaminase gene (PMID: 29945998). Therefore, even considering that no mice show complete regression in orthotopic models, the authors should analyze whether, at the experimental endpoints, tumor masses still contain replicating and infectious viral particles and whether infected cells remain sensible to 5-FC treatment. Moreover, they could analyze whether viral particles spread evenly throughout the entire tumor mass reaching also migrating tumor cells.

-          Authors should add, throughout the result section of the manuscript, the number of animals used in the in vivo experiments

-          Authors should add, throughout the result section of the manuscript, how many times they repeated the experiments, since sometimes it is not clear. Just as an example, they stated that they cultured cells in triplicate to perform the cytotoxicity assay meaning, I suppose, they analyze 3 wells per condition at a time, but they do not say how many times the experiment was repeated and the lack of error bars in some of the data in Figure2 does not help to understand the variability of the experiment and suggests that the experiment was done only once. If this is the case, I suggest repeating the experiment and merging all the data into a new figure.

-          I suggest to add in figure 1 at least one example of the cytofluorimetric data (as gating strategies in FSC/SSC and fluorescence intensity dot blots) they obtained and not just aggregate results in a graph. Even in this case, the lack of error bars suggests that the experiment was done only once, but they do not clearly state that.

-           Authors should describe how and when they transduced the cells for the subcutaneous experiments. Do they infect cells in vitro just before injecting them? Do they inject the virus and the cells at the same time or at different times?

-          The p-values in lines 287,288,300 and 304 are correct or swapped? Since it seems they have a bigger effect with murine than human cells.

-          In Figures 5 and 7, data should be presented with Log-scale on the Y-axis to make them easier to read. Moreover, the authors should add the numeric data of figure % in the text as they did for figure 7. As suggested before, for figure 7 authors should add the number of times they repeated the experiments.

-          Authors found about 10-20 RRV copies/100 cells in lymphohematopoietic tissues in the immunodeficient subcutaneous tumor model. I do not think these values represent, as they stated, “low levels”. Considering they use a replicating virus, the infection level could increase with time especially after completing the cytotoxic 5-FC treatment. Authors should comment better this point.

-          Authors should indicate also in the result section that for the orthotopic model they injected RRV viruses 2 days after tumor injection.